# Cross-Talk of Toll-Like Receptor 5 and Mu-Opioid Receptor Attenuates Chronic Constriction Injury-Induced Mechanical Hyperalgesia through a Protein Kinase C Alpha-Dependent Signaling

**DOI:** 10.3390/ijms22041891

**Published:** 2021-02-14

**Authors:** Ching Chang, Hung-Kai Liu, Chao-Bin Yeh, Ming-Lin Yang, Wen-Chieh Liao, Chiung-Hui Liu, To-Jung Tseng

**Affiliations:** 1Department of Anatomy, School of Medicine, Chung Shan Medical University, 40201 Taichung, Taiwan; sa10940134@gmail.com (C.C.); ultimateken666@gmail.com (H.-K.L.); mly4736@csmu.edu.tw (M.-L.Y.); khrnangel@gmail.com (W.-C.L.); chiunghui.liu@gmail.com (C.-H.L.); 2Department of Emergency Medicine, Chung Shan Medical University Hospital, 40201 Taichung, Taiwan; sky5ff@gmail.com; 3Department of Emergency Medicine, School of Medicine, Chung Shan Medical University, 40201 Taichung, Taiwan; 4Department of Medical Education, Chung Shan Medical University Hospital, 40201 Taichung, Taiwan

**Keywords:** chronic constriction injury, mechanical hyperalgesia, toll-like receptor 5, FLA-ST ultrapure, mu-opioid receptor, protein kinase Cα

## Abstract

Recently, Toll-like receptors (TLRs), a family of pattern recognition receptors, are reported as potential modulators for neuropathic pain; however, the desired mechanism is still unexplained. Here, we operated on the sciatic nerve to establish a pre-clinical rodent model of chronic constriction injury (CCI) in Sprague-Dawley rats, which were assigned into CCI and Decompression groups randomly. In Decompression group, the rats were performed with nerve decompression at post-operative week 4. Mechanical hyperalgesia and mechanical allodynia were obviously attenuated after a month. Toll-like receptor 5 (TLR5)-immunoreactive (ir) expression increased in dorsal horn, particularly in the inner part of lamina II. Additionally, substance P (SP) and isolectin B4 (IB4)-ir expressions, rather than calcitonin-gene-related peptide (CGRP)-ir expression, increased in their distinct laminae. Double immunofluorescence proved that increased TLR5-ir expression was co-expressed mainly with IB4-ir expression. Through an intrathecal administration with FLA-ST Ultrapure (a TLR5 agonist, purified flagellin from *Salmonella* Typhimurium, only the CCI-induced mechanical hyperalgesia was attenuated dose-dependently. Moreover, we confirmed that mu-opioid receptor (MOR) and phospho-protein kinase Cα (pPKCα)-ir expressions but not phospho-protein kinase A RII (pPKA RII)-ir expression, increased in lamina II, where they mostly co-expressed with IB4-ir expression. Go 6976, a potent protein kinase C inhibitor, effectively reversed the FLA-ST Ultrapure- or DAMGO-mediated attenuated trend towards mechanical hyperalgesia by an intrathecal administration in CCI rats. In summary, our current findings suggest that nerve decompression improves CCI-induced mechanical hyperalgesia that might be through the cross-talk of TLR5 and MOR in a PKCα-dependent manner, which opens a novel opportunity for the development of analgesic therapeutics in neuropathic pain.

## 1. Introduction

Neuropathic pain is a neurological disorder caused by dramatic physical injury to peripheral or central primary afferents which disturbs life quality extensively [1,2]. The clinical pain symptoms, including hyperalgesia (noxious stimuli), allodynia (innocuous stimuli), paresthesia (a prickling or burning sense) and dysesthesia (deficiency of touch sensitivity), have been identified [3,4]. Central sensitization is defined as the pre-synaptic terminals increase in excitability or decrease in inhibition that modulate the post-synaptic neurons after peripheral nerve injury [5,6]. Accordingly, the pre-synaptic terminals in the distinct laminae of dorsal horn are critical to establish the synaptic plasticity of the spinal cord [6,7]. More notably, unmyelinated C and myelinated Aδ terminals have been distinguished as the peptidergic (substance P (SP) and calcitonin gene-related peptide (CGRP)) and non-peptidergic (isolectin B4 (IB4)) terminals histochemically, recognized as the nociceptors [8,9,10]. Thus, the pre-clinical rodent models such as chronic constriction injury (CCI), spared nerve injury (SNI) and lumbar 5 spinal nerve ligation (SNL) are established for investigating potential mechanisms in central sensitization [11,12,13].

Toll-like receptors (TLRs), a family of pattern-recognition receptors, mediate the innate immunity in response to pathogen associated molecular patterns (PAMPs) or danger associated molecular patterns (DAMPs) [14,15,16]. Cell membrane TLRs generally branches out the signaling, based on two distinct adapter proteins in innate immune cells [14,17,18]. The myeloid differentiation primary response protein88 (MyD88)-dependent signaling results in the activation of nuclear factor kappa-light-chain-enhancer of activated B cells and mitogen-activated protein kinases (MAPKs) for the production of pro-inflammatory cytokines [14]. By contrast, the MyD88-independent signaling leads to interferon production by the activation of Toll/interleukin-1 receptor -domain-containing adapter-inducing interferon-β [16,17]. More remarkably, TLRs express in dorsal root ganglion (DRG) neurons and even to primary afferents in the dorsal horn of the spinal cord and the dermis of the glabrous skin [19]. The specific role of TLRs in DRG neurons is mainly emphasized on the induction of inflammatory and neuropathic pain, in light of the events of TLRs signaling in immune cells [20,21].

Opioid receptors (ORs) belong to the superfamily of G protein-coupled receptors (GPCRs), the members of which modulate G protein for activating signal transduction in opioid analgesia [22,23,24]. For instance, Gα and Gβγ subunits of mu-OR (MOR) are dissociated when MOR is activated by endogenous opioid peptide (endorphins) or exogenous opioid agonists (morphine and DAMGO ([D-Ala2,N-Me-Phe4,Gly5-ol]enkephalin)), successively acting on different intracellular signaling. Gαq subunit activates phospholipase C to produce inositol trisphosphate and diacylglycerol for the phosphorylation of protein kinase C (PKC) [22,23]. Gαi/o subunit inhibits the adenylate cyclase for producing cyclic adenosine monophosphate, which inactivates protein kinase A (PKA) [23,25]. Morphine and DAMGO further mediate MAPKs activation that has been directly verified by the use of its signaling inhibitors, such as PKC or PKA inhibitors, in MOR-dependent signaling [23]. Moreover, DAMGO mediates MOR desensitization through the phosphorylation of GPCR kinases to promote β-arrestin binding, leading to rapid receptor internalization and re-sensitization, also activates MAPKs [23,26].

To date, mechanisms of receptor interactions between OR and OR/other classes of GPCRs give the hypothesis of cross-talk that occurs in heterodimer formation, heterologous desensitization and intracellular signaling [24,25,26,27,28,29]. For example, CYM51010, a MOR-DOR-biased agonist, increases co-expression of MOR and DOR in injured DRG neurons and has analgesic effects on neuropathic pain by subcutaneous administration [27,28]. The existence of heterodimer formations between MOR and DOR provides a further inspiration and emphasizes a modified signaling regulation of their therapeutic effects [24,27,29]. For instance, DOR-mediated MAPKs activation requires PKCβr activity for activating Ras signaling that is different from MOR-mediated MAPKs activation in DRG neurons. More importantly, cross-talk between MOR and cannabinoid receptor 1 (CBR1) has been proved that administration with δ 9-tetrahydrocannabinol, a CBR1 agonist, enhances the morphine effectiveness and leads to a significant reduction of MAPKs activation [30,31].

Due to the reliable efficacy of nerve decompression in the improvements of CCI-induced pain hypersensitivity in Sprague-Dawley rats, our existing purposes are setup to (1) interpret the variation of TLR5 expression and its interrelation with the changes of synaptic plasticity in dorsal horn; (2) illustrate the potential role of TLR5 in mechanical hyperalgesia and mechanical allodynia by an intrathecal administration with FLA-ST Ultrapure (flagellin, a TLR5 agonist); (3) elucidate MOR expression in dorsal horn and explain whether its signaling mechanisms are in association with the protein kinases (PKC and PKA); and (4) recognize whether TLR5 can regulating MOR/PKC signaling by the mechanism of cross-talk for attenuating CCI-induced mechanical hyperalgesia and mechanical allodynia.

## 2. Results

### 2.1. Nerve Decompression Efficiently Relieved CCI-Induced Pain Hypersensitivity

To examine the influence of nerve decompression in relieving pain hypersensitivity after CCI, we conducted behavioral assessments and compared the difference between CCI and Decompression groups (Figure 1). In CCI group, the decreases of withdrawal threshold were revealed on ipsilateral sides, indicating mechanical hyperalgesia, from post-operative week (POW) 2 to POW 8 (F_(3,46)_ = 259.9; *p* < 0.0001) (Figure 1A). More importantly, in Decompression group, withdrawal thresholds showed on ipsilateral sides have no considerable difference with that on contralateral sides at POW 8 (*p* = 0.6066) (Figure 1A). Likewise, the reductions of mechanical threshold on ipsilateral sides were shown in CCI group during the entire experimental period (F_(3,46)_ = 494.8; *p* < 0.0001) (Figure 1B). At POW 8 in Decompression group, mechanical thresholds on ipsilateral sides revealed similar outcomes as which measured on contralateral sides (*p* = 0.4158) (Figure 1B).

### 2.2. CCI-Induced Decrease of TLR5 Expression in Dorsal Horn Was Reversed by Nerve Decompression

To recognize whether the synaptic plasticity of TLR5 changed as a result of nerve decompression in CCI rats, we studied the TLR5-immunoreactive (ir) expression in dorsal horn through immunohistochemistry (Figure 2). On contralateral sides, TLR5-ir expression exhibited a dense dot-like appearance mainly in the inner part of lamina II, in both CCI and Decompression groups (Figure 2A,C,E). At POW 2, the significant loss of TLR5-ir expression in dorsal horn was showed on ipsilateral sides either in CCI group or Decompression group (Figure 2B). Comparatively, ipsilateral sides revealed a more obvious increase of TLR5-ir expression in Decompression group than that in CCI group at POW 8 (Figure 2D,F).

We verified the temporal changes of TLR5-ir expression by quantitation of TLR5-ir dorsal horn areas in the medial and lateral portions of dorsal horn (Figure 3). In CCI group, the medial portion of dorsal horn on ipsilateral sides exhibited a large reduction of TLR5-ir dorsal horn areas through the end of experiments (F_(3,16)_ = 198.1; *p* < 0.0001) (Figure 3A). More importantly, at POW 8 in Decompression group, values of TLR5-ir dorsal horn areas on ipsilateral sides increased to the level of close to those on contralateral sides in the medial portion of dorsal horn (*p* = 0.0229) (Figure 3A). However, the TLR5-ir dorsal horn areas in the lateral portion of dorsal horn showed the similar pattern between both groups during the entire experimental period (*p* > 0.05, respectively) (Figure 3B).

### 2.3. Nerve Decompression Modulated the Reversal of Synaptic Plasticity in Dorsal Horn

To recognize the influence of nerve decompression in CCI-induced changes of synaptic plasticity in dorsal horn, we applied immunohistochemical staining with antiserums against SP, CGRP and IB4 (Figure 4). SP is known as a marker of unmyelinated C terminals. On contralateral sides, SP-ir expression showed beaded particles mostly in lamina I and extended to the outer part of lamina II (Figure 4A). At POW 8, partial loss of SP-ir expression in the equivalent laminae was detected on ipsilateral sides in CCI group (Figure 4B). Also, on the ipsilateral sides, Decompression group showed more abundant SP-ir expression in the corresponding laminae (Figure 4C). Similarly, on contralateral sides, peptidergic C and Aδ terminals were illustrated by CGRP-ir expression, which showed dense varicosities in lamina I, the outer part of lamina II and even lamina V (Figure 4D). However, on the ipsilateral sides, there was not a noticeable difference in CGRP-ir expression at POW 8 between CCI and Decompression groups (Figure 4E,F). Besides, IB4 is also known as a marker of unmyelinated C terminals. IB4-ir expression on contralateral sides which exhibited irregular structures expressed generally in the inner part of lamina II (Figure 4G). On ipsilateral sides, a reduction of the amount of IB4-ir expression was observed at POW 8 in CCI group (Figure 4H). In contrast, an observable increase of IB4-ir expression was revealed on the ipsilateral sides in Decompression group (Figure 4I).

These morphological evidences were quantified as dorsal horn areas in the medial portion of dorsal horn on contralateral and ipsilateral sides independently (Figure 5). As the results demonstrated, SP-ir dorsal horn area revealed a significant increase in Decompression group at POW 8 (vs. CCI group, *p* < 0.0001) (Figure 5A). At the same time point, CGRP-ir dorsal horn area in Decompression group were similar to those in CCI group (*p* = 0.9513) (Figure 5B). Importantly, a noticeable increase of IB4-ir dorsal horn area was also detected in Decompression group at POW 8 (vs. CCI group, *p* < 0.0001) (Figure 5C).

### 2.4. Increase of TLR5 Expression Was Predominantly Observed Which Co-Expressed with IB4 Expression after Nerve Decompression

In order to determine the actual TLR5-ir expression in dorsal horn, double immunofluorescence was applied to evaluate its co-expressions with peptidergic or non-peptidergic terminals (Figure 6). At POW 8, TLR5- and SP-ir expressions were not meaningfully co-expressed in both groups in lamina I (Figure 6A,B). Quantitative dorsal horn areas demonstrated that there were about 3.86% co-expression in CCI group and 8.84% co-expression in Decompression group (Figure 6C). Moreover, it showed that TLR5- and CGRP-ir expressions in both groups co-expressed around the border between the outer and inner part of lamina II (Figure 6D,E). However, only about 10.00% TLR5-ir expression was co-expressed with CGRP-ir expressions in Decompression group (Figure 6F). It is surprising that TLR5-ir expression almost co-expressed with IB4-ir expression mainly in the inner part of lamina II in CCI group (Figure 6G). This phenomenon of co-expression between TLR5- and IB4-ir expressions was also detected significantly in Decompression group (Figure 6H). Quantitative dorsal horn areas showed that TLR5-ir expressions mostly co-expressed with IB4-ir expressions (about 97.11% in CCI group and 94.08% in Decompression group) (Figure 6I). In summary, these above findings confirmed the facts in Decompression group that increase of TLR5-ir expression significantly localized in non-peptidergic terminals, which were unmyelinated C terminals, in dorsal horn.

### 2.5. FLA-ST UtrapureAttenuated CCI-Induced Mechanical Hyperalgesia by an Intrathecal Administration in a Dose-Responsive Manner

To explore whether the TLR5 contributes to an analgesic effect on CCI-induced pain hypersensitivity in chronic pain state, we evaluated the CCI rats after an intrathecal administration with FLA-ST Ultrapure at POW 8 (Figure 7). After CCI, difference in withdrawal thresholds and mechanical thresholds at POW 8 were defined as the baseline values at PIH 0 and applied for the studies of pharmacological intervention. FLA-ST Ultrapure affected the difference in withdrawal threshold (F_(12,40)_ = 6.47; *p* < 0.0001) in a dose-responsive manner (F_(3,40)_ = 29.13; *p* < 0.0001) and it altered the temporal pattern of mechanical hyperalgesia (F_(4,40)_ = 51.78; *p* < 0.0001) after an intrathecal administration (Figure 7E). The 0.9 μg FLA-ST Ultrapure group almost reversed CCI-induced difference in withdrawal thresholds to normal values from PIH 2 to PIH 6 and returned to the baseline level at PIH 24 (F_(4,10)_ = 47.84; *p* < 0.0001). In the 0.3 μg FLA-ST Ultrapure group, difference in withdrawal thresholds also intensified up rapidly, comparing to the pattern of the 0.9 μg FLA-ST Ultrapure group (F_(4,10)_ = 17.20; *p* = 0.0002). As well, difference in withdrawal thresholds began to reverse at PIH 2 and gradually declined from PIH4 to PIH6 that was observed in the 0.1 μg FLA-ST Ultrapure group (F_(4,10)_ = 12.19; *p* = 0.0007). In Vehicle group, these difference in withdrawal thresholds were examined at the each time points (F_(4,10)_ = 0.2787; *p* = 0.8851), which were used for comparing the influence of an intrathecal administration.What is more, there was not any effects of FLA-ST Ultrapure on difference in mechanical thresholds (F_(12,40)_ = 0.25; *p* = 0.9940) showed with any concentrations (F_(3,40)_ = 0.29; *p* = 0.8293) and the time-based mode of mechanical allodynia (F_(4,40)_ = 0.76; *p* = 0.5559) did not change after an intrathecal administration (Figure 7F). For instance, difference in withdrawal thresholds were not affected in the 0.9 μg FLA-ST Ultrapure group (F_(4,10)_ = 0.46; *p* = 0.7637). Taken together, depending on the temporal changes of difference in withdrawal thresholds, we proved that FLA-ST Ultrapurehas positive effects on mechanical hyperalgesia in a dose-responsive manner. In comparison, in all the FLA-ST Ultrapure groups, there was only a slight change of difference in mechanical thresholds at each time points, representing that TLR5 were not involved in the modulation of mechanical allodynia.

### 2.6. Nerve Decompression Induced the Increase of MOR Expression, Whereas Its Co-Expression Was Mainly Detected with IB4 Expression

The sections of the spinal cord demonstrated the potential MOR-ir expression through immunostaining with antiserum against MOR (Figure 8A,C,D). The standard pattern of MOR-ir expression on contralateral sides revealed abundant dense particles, which mainly expressed in lamina II (Figure 8A). Reduction of MOR-ir expression on ipsilateral side was significantly detected at POW 8 in CCI group (Figure 8C). At the same time point, an obvious increase of MOR-ir expression was revealed on ipsilateral side in Decompression group (Figure 8D). In both groups, these MOR-ir expressions were quantified as dorsal horn area in the medial portion of dorsal horn on contralateral and ipsilateral sides separately (Figure 8B). Notably, at POW 8 in Decompression group, the ipsilateral sides exposed a substantial increase of MOR-ir dorsal horn area (vs. CCI group, *p* < 0.0001) (Figure 8B).

For the purpose of evaluating the actual MOR-ir expression in dorsal horn, we conducted double immunofluorescence to illustrate its co-expressions with peptidergic and non-peptidergic terminals (Figure 9). At POW 8 in Decompression group, MOR-ir expression co-expressed slightly with SP-ir expression around the border between lamina I and II (Figure 9A). MOR-ir expression exhibited partial co-expression with CGRP-ir expression in the outer part of lamina II, similar to those co-expressing pattern of SP-ir expression (Figure 9B). Furthermore, MOR- and IB4-ir expressions were nearly co-expressed in their correspondent region, mainly in lamina II (Figure 9C). Quantitative dorsal horn areas in Decompression group revealed that about 97.16% MOR-ir expression co-expressed with IB4-ir expression (Figure 6D). Taken together, these above evidence confirmed that MOR-ir expression also co-expressed with non-peptidergic terminals in lamina II noticeably.

### 2.7. Increase of pPKCα Expression, Rather Than pPKA RII Expression, in Dorsal Horn, Where Its Co-Expression Was Mostly Observed with IB4 Expression after Nerve Decompression

To identify the potential intracellular signaling, the sections of the spinal cord were immunostained with antiserum against pPKCα and pPKA RII (Figure 10). On contralateral sides, pPKCα-ir expression formed a dotted appearance scattered mainly in the inner part of lamina II (Figure 10A). At POW 8 in CCI group, a lesser amount of pPKCα-ir expression was detected on ipsilateral sides (Figure 10B). In contrast, a noticeable increase of pPKCα-ir expression was observed on ipsilateral sides in Decompression group (Figure 10C). In comparison, contralateral sides exhibited the pPKA RII-ir expression was showed in deep laminae (Figure 10D). Nevertheless, on the ipsilateral sides, there was no obvious difference in pPKA RII-ir expression at POW 8, between CCI and Decompression groups (Figure 10E,F). Dorsal horn areas were quantified in the medial portion of dorsal horn on contralateral and ipsilateral sides separately (Figure 11). At POW 8, the results in Decompression group had demonstrated a considerable increase of pPKCα-ir dorsal horn area (vs. CCI group, *p* < 0.0001) (Figure 11A). At the same time point, pPKA RII-ir dorsal horn area in Decompression group were parallel to that in CCI group (*p* = 0.8055) (Figure 11B).

For the purpose of understanding the real pPKCα- and pPKA RII-ir expressions in dorsal horn, we evaluated their co-expressions in peptidergic and non-peptidergic terminals by double immunofluorescence (Figure 12). At POW 8 in Decompression group, an obvious pPKCα-ir expression was detected in the medial portion of dorsal horn which showed some co-expressions with SP- or CGRP-ir expression in lamina I and near the border between the outer and inner part of lamina II (Figure 12A,C). Furthermore, pPKCα- and IB4-ir expressions were almost co-expressed in their equivalent region, particularly in the inner part of lamina II (Figure 12E). It is unexpected that pPKA RII-ir expression showed more noticeable co-expressions with IB4-ir expression than with SP- or CGRP-ir expression (Figure 12B,D,F). Quantitative dorsal horn areas in Decompression group showed that about 92.08% pPKCα-ir expression co-expressed with IB4-ir expression (Figure 12G). However, only about 67.72% pPKA RII-ir expression co-expressed with IB4-ir expression in Decompression group (Figure 12H). In summary, these above findings proved that pPKCα-ir expression also expressed in non-peptidergic terminals significantly.

### 2.8. Go 6976Re-Induced FLA-ST Ultrapure- and DAMGO-Mediated Reversal of Mechanical Hyperalgesia by an Intrathecal Administration Dose-Dependently

To explore whether PKCα contributes to TLR5- or MOR-mediated attenuation of pain hypersensitivity in a chronic pain state, we evaluated the CCI rats after an intrathecal administration with FLA-ST Ultrapure or DAMGO and in combined with Go 6976 at POW 8 (Figure 13). Go 6976 affected the FLA-ST Ultrapure-induced difference in withdrawal threshold (F_(12,40)_ = 6.03; *p* < 0.0001) in a dose-responsive manner (F_(3,40)_ = 33.95; *p* < 0.0001) and changed the temporal pattern of reversed mechanical hyperalgesia (F_(4,40)_ = 20.44; *p* < 0.0001) after an intrathecal administration (Figure 13A). Moreover, Go 6976 disturbed the DAMGO-induced difference in withdrawal threshold (F_(12,40)_ = 5.95; *p* < 0.0001) dose-dependently (F_(3,40)_ = 32.69; *p* < 0.0001) and altered the time-based mode of reversed mechanical hyperalgesia (F_(4,40)_ = 17.28; *p* < 0.0001) after an intrathecal administration (Figure 13B). However, Go 6976 did not have any effects on difference in mechanical thresholds (F_(12,40)_ = 0.32; *p* = 0.9817 for FLA-ST Ultrapure; F_(12,40)_ = 0.16; *p* = 0.9993 for DAMGO) and show with any concentrations (F_(3,40)_ = 0.25; *p* = 0.8586 for FLA-ST Ultrapure; F_(3,40)_ = 0.04; *p* = 0.9877 for DAMGO) (Figure 13C,D). Go 6976 did not alter DAMGO-induced temporal pattern of reversed mechanical allodynia after an intrathecal administration (F_(4,40)_ = 191.4; *p* < 0.0001) (Figure 13D). In summary, based on FLA-ST Ultrapure- or DAMGO-induced reversal of difference in withdrawal thresholds, we showed that Go 6976 has significant effects on the re-induction of mechanical hyperalgesia in a dose-responsive manner.

## 3. Discussion

Growing evidence in CCI rats demonstrate the changes of synaptic plasticity in dorsal horn [10,32,33]. For example, SP expression decreased in lamina I and the outer part of lamina II, while it also shows a significant decrease of CGRP expression in lamina I, lamina II and deeper lamina V. Our previous study proves that nerve decompression can reverse CCI-induced decrease of SP expression but did not affect CGRP expression, in dorsal horn [10]. In current reports, we further illustrated that a noticeable decrease of IB4 expression was detected in the inner part of lamina II in CCI group. Importantly, nerve decompression provided a possibility for the reversal of IB4 expression in its corresponding lamina. Comparable results show that both of SP and IB4 expressions has an equivalent pattern of decreasing trend in their distinct laminae but these expressions are reversed within one month after CCI, not after SNI and crush [33]. According to our observations, nerve decompression reversed the SP and IB4 expressions, implying the unmyelinated C terminals in dorsal horn were related to the relief of CCI-induced pain hypersensitivity. Significantly, a recent study reveals that an intrathecal administration with IB4-saporin depletes IB4 and ATP-gated ionotropic purinergic receptor 3 (P2X3) in dorsal horn, which attenuates carrageenan-induced inflammatory pain in mice [34]. Consequently, nerve decompression in CCI rats provides the opportunity for studying the mechanisms of central sensitization, especially the re-establishment of synaptic plasticity in dorsal horn.

Numerous members of TLRs, including cell membrane (TLR1, TLR2, TLR4, TLR5 and TLR6) and endosomal (TLR3, TLR7 and TLR9) TLRs, have been identified in mouse DRG neurons [14,16]. Most important of all, mouse primary cell cultures expose that protein and mRNA levels of TLR5 are detected in DRG neurons [35]. Besides, TLR5 expression is also observed in DRG neurons, where it co-expresses with neurofilament 200 (NF-200) expression both in mice and human [19]. It further indicates that myelinated Aβ terminals in the deep laminae of dorsal horn, which is identical with those in the dermis of the glabrous skin, show the TLR5 expression in mice. In this study, our results indicated that decrease of TLR5-ir expression in CCI group was detected mainly in the inner part of lamina II, especially in the medial portion of dorsal horn, whereas the TLR5-ir expression increased in the corresponding lamina in Decompression group. By double immunofluorescence, TLR5-ir expression in both groups almost co-expressed with IB4-ir expression in the inner part of lamila II. Our study therefore emphasized that TLR5 expression expressed in unmyelinated C terminals, especially non-peptidergic terminals, in dorsal horn.

The cell membrane TLRs are well-characterized receptors for the innate immunity that contributes to the recognition of various PAMPs, including lipopeptides (for TLR2), LPS (for TLR4) and flagellin (for TLR5) [14,20,36]. Resent evidences have intensified that flagellin-induced TLR5 up-regulation in immune cells potentially participates in the production of neuropathic and inflammatory pain [35,37,38]. Nevertheless, it is uncertain that whether TLR5 up-regulation in DRG neurons and their nerve terminals also function as the potential contributors. In our report, the results supported that FLA-ST Ultrapure attenuated mechanical hyperalgesia in a dose-responsive manner by an intrathecal administration in CCI group. But this is not consistent with what had been found in mice [19]. For instance, flagellin induces mechanical allodynia by an intraplantar administration. Based on the first observation of intrathecal administration with TLR5 agonist, our finding raised another possibility that FLA-ST Ultrapure resulted in TLR5 up-regulation for attenuating CCI-induced mechanical hyperalgesia. Furthermore, the noxious endogenous ligands have been identified as DAMPs, such as high-mobility group box protein 1 (HMGB1), heat shock proteins, S100 proteins, nucleic acids and histone proteins, which are released during tissue injury [38,39]. For example, an intraplantar administration with HMGB1 induces acute mechanical allodynia, which is attenuated by a co-treatment with TH1020, a TLR5 antagonist, in Sprague-Dawley rats. While these novel insights have been a desired outcome, it is necessary to verify which the endogenous ligands of TLR5 in dorsal horn are involved in the relief of pain hypersensitivity after CCI.

MOR and delta-OR (DOR) regulates the release of SP from the pre-synaptic terminals to bind to post-neuronal neurokinin 1 receptor in dorsal horn for transmitting nociception [8,40,41]. Therefore, a significant reduction of MOR is detected in dorsal horn after CCI, which is compared to those after SNI and SNL [42,43,44]. Our previous laboratory data also illustrate that an observable decrease of MOR, which is parallel to that of DOR, is detected in dorsal horn after CCI, whereas they dramatically increased after nerve decompression [8]. In current study, we further observed that this increased MOR-ir expression expressed in lamina II in Decompression group, which co-expressed mainly with IB4-ir expression. What is more, various PKC isoforms which are known as the secondary messengers are involved in GPCRs signaling for the establishment of peripheral and central sensitization [45,46,47,48]. For example, PKCα activation can modulate AMPA receptors in dorsal horn for mediating Complete Freund’s adjuvant (CFA)-induced inflammatory pain. However, our current study suggested that increase of pPKCα-ir expression in Decompression group mostly co-expressed with IB4-ir expression in lamina II. In summary, we strongly considered that non-peptidergic terminals might up-regulate MOR to activate PKCα-dependent signaling for attenuating pain hypersensitivity.

Well-known MOR agonists, such as morphine, endomorphin and DAMGO, provide some information for the roles of opioid analgesia in attenuating several types of pain hypersensitivity in CCI rats [8,40,49]. For example, DAMGO diminishes CCI-induced thermal hyperalgesia, cold allodynia and mechanical allodynia by an intrathecal administration. Our earlier findings also prove that both DAMGO and SNC80 (DOR agonist) administrated intrathecally can attenuate CCI-induced thermal hyperalgesia and mechanical allodynia dose-dependently [8]. In contrasts, a previous report shows that calphostin C, a PKC inhibitor, can reverse DAMGO-mediated analgesia by the intrathecal and intraplantar administration [50]. More interestingly, several types of cross-talk between ORs and TLR4 are considered to have a potential role in opioid analgesia and immune function [51,52,53]. For instance, morphine-mediated ORs and TLR4 cross-talk inhibits the LPS-induced secretion of TNFα that is due to the complex formation of the β-arrestin 2/TNF receptor-associated factor 6 in mast cells. Furthermore, only the co-treatment with MOR and DOR antagonists can abolish this morphine-induced inhibition. Accordingly, the “agonist-selective” (or ligand-biased) theory was suggested to provide the possible explanations that endogenous opioids might act as flagellin to activate TLR5 for relieving CCI-induced mechanical hyperalgesia after nerve decompression. Remarkably, TLR5 up-regulation can induce PKCα activation directly, which moderates L. pneumophila-related pro-inflammatory production in human lung epithelium [18,54]. Based on this observation, we confirmed that Go 6976 has dose-dependent effects on the re-induction of FLAST Ultrapure- and DAMGO-mediated reversal of CCI-induced mechanical hyperalgesia. We therefore suggested that PKCα activity contributes to mediating TLR5 and MOR signaling for attenuating mechanical hyperalgesia after nerve decompression in CCI rats. Nevertheless, how the TLR5/PKCα signaling directly regulates down-stream molecules or inhibits excitatory receptor activity for attenuating CCI-induced mechanical hyperalgesia after nerve decompression remains unresolved.

## 4. Materials and Methods

### 4.1. Animals

Adult male Sprague-Dawley rats weighing 200–250 g used in these experiments were placed in a temperature- and humidity-controlled room with a 12 h light/dark cycle. Food and water were available ad libitum. All the procedures were conducted in accordance with the ethical guidelines set up by the International Association for the Study of Pain (IASP) on the use of laboratory animals in the experimental research and the protocol (project identification code: 2242, 23 September 2019) was approved by the Animal Committee of Chung Shan Medical University, School of Medicine, Taichung, Taiwan [55] (IASP Committee, 1980).

### 4.2. Surgery

CCI was performed in rats following the established surgical procedures [11]. Briefly, under pentobarbital anesthesia (60 mg/kg, intraperitoneal administration), the right sciatic nerve was exposed at mid-thigh level by freeing the adhering fascia between the gluteus and biceps femoris muscles. Four ligatures with 4/0 chromic gut (Johnson & Johnson, Mumbai, India) were tied loosely around the sciatic nerve at 1-mm intervals above the nerve’s trifurcation. The ligatures constricted only about 1/3–1/4 of the diameter of sciatic nerve and produced a brief twitch in the muscle around the exposure site. In this study, the surgical side was defined as the ipsilateral side and its control side was defined as the contralateral side in the following analyses.

To examine the effects of nerve decompression in CCI model on pain hypersensitivity and TLR5-ir expression in dorsal horn, the rats after CCI were randomly assigned into either CCI group or Decompression group as described previously [10]. Before nerve decompression, all four ligatures were visible although reactive fibrosis became prominent at POW 4. Then, ligatures could be untied and removed without destroying the surrounding tissues under dissecting microscope (Olympus, Venter Valley, PA, USA), which was defined as Decompression group. In CCI group, all the ligatures were remained and reserved throughout the experimental period.

### 4.3. Behavioral Assessments

#### 4.3.1. Mechanical Hyperalgesia

Mechanical hyperalgesia is evaluated by the noxious pinprick stimulation (A von Frey–type 0.5 mm filament) with Dynamic Plantar Aesthesiometer (Code: 37450, Ugo Basile, Comerio-Varese, Italy) [56]. Minimal force (g) is automatically measure as the time elapsed from the onset of rounded tip stimulation to the withdrawal of hindpaw. Each hindpaw is alternatively tested seven times with a minimal interval of 5 min between measurements. The values of last five consecutive measurements are used for the analysis and average as a withdraw threshold. To elucidate the decreases of withdraw threshold, the values of ipsilateral side subtracting that of contralateral side are defined as the difference in withdrawal threshold.

#### 4.3.2. Mechanical Allodynia

Mechanosensitivity was determined by measuring the withdrawal thresholds with a series of calibrated von Frey filaments (Senselabaesthesiometer, Somedic Sales AB, Stockholm, Sweden) according to an up-and-down method [8]. The examiner touched the plantar surface of the hindpaw with a filament until the bending angle reached 45° and a brisk withdrawal or paw flinching was noted, which was considered as a positive response. Hence, the value of mechanical threshold was defined as the minimal force (g) applied by a von Frey filament which initiated a positive withdrawal response of the hindpaw. The difference in mechanical threshold was also defined by the values of ipsilateral side subtracting that of contralateral side.

### 4.4. Immunohistochemistry

At the end of all experiments, the rats were deeply anesthetized using pentobarbital (100 mg/kg, i.p.) and sacrificed by intracardiac perfusion of 4% paraformaldehyde in 0.1 M phosphate buffer (PB) at pH 7.4. After perfusion, the spinal cords were removed and further immersed in fixative for additional 6 h before shifted to 0.1 M PB for storage. Prior to sectioning, samples were rinsing thoroughly, cryoprotected with 30% sucrose in 0.1 M PB overnight. Then the spinal cord was cut at a plane perpendicular in a thickness of 50 μm per section with a sliding microtome (HM440E; Microm, Walldorf, Germany), labeled sequentially and stored at −20 °C. The sections for immunohistochemistry were treated with 0.5% Triton X-100 in 0.5 M Tris buffer (Tris) at pH 7.6 for 30 min and processed for immunostaining. Briefly, the sections were quenched with 1% H_2_O_2_ in methanol, blocked with 5% normal goat serum in 0.5% nonfat dry milk/Tris and then incubated with primary antiserum at 4 °C overnight. All of these primary antisera against including: (1) rabbit polyclonal TLR5 (1:200, 19810-1-AP, Proteintech Group, Inc., Chicago, IL, USA); (2) rabbit polyclonal SP (1:1000, AB1566, Millipore, Billerica, MA, USA); (3) rabbit polyclonal CGRP (1:1000, AB15360, Millipore, Billerica, MA, USA); (4) biotinylated IB4 (1:1000, L 2140, Sigma, UK); (5) rabbit polyclonal pPKCα (1:200, 1195-1,Epitomics, Burlingame, CA, USA); (6) rabbit polyclonal pPKA RII (1:200, 1151-1, Epitomics, Burlingame, CA, USA) and (7) rabbit polyclonal MOR (1:1000, RA10104, Neuromics, Edina, MN, USA) respectively. After rinsing in Tris, the sections were incubated with biotinylated goat anti-rabbit IgG (1:100; Jackson ImmunoResearch Laboratories, West Grove, PA, USA) for 1 h, except the biotinylated IB4 treatment. They were then incubated with avidin-biotin complex horseradish peroxidase reagent (Vector Laboratories, Burlingame, CA, USA) for another hour and the reaction products were demonstrated with 3,3′-diaminobenzidine (DAB, Sigma–Aldrich Co., St. Louis, MO, USA).

### 4.5. Imaging Analysis

Areas with ir expressions in dorsal horn were quantified following a protocol from previous published methods [10]. Superficial dorsal horn containing laminae I and II was demonstrated as a dark band under an Olympus microscope (BH2; Olympus, Tokyo, Japan) equipped with a dark-field apparatus and a blue filter at a magnification of 100×. Without changing the position of the slide, the same field was photographed under bright-field condition. The areas occupied by ir expressions within the boundary on the superimposed image of these two photographs were considered to be ir expressions in dorsal horn for morphometric analysis.

The principle of calculating ir expressions in dorsal horn was based on the marked differences in the optical densities between the ir expression and the background measured with Adobe Photoshop Elements 2.0 (San Jose, CA, USA). Except for measuring the optical densities of the defined ir expression and the background, no any processing of images was performed with this software. Optical density of dorsal column was defined as the background optical density of that section, which was compared with the optical density of ir expression in dorsal horn. Numbers of ir expression on each side of the spinal cord and the areas of dorsal horn were measured as ir expression (μm^2^) for each side of the spinal cord. Importantly, the medial part of dorsal horn was corresponding to the major receptive field of the sciatic nerve; thus, we further focused on the changes of ir expression in that part in following imaging analysis.

### 4.6. Double Immunofluorescence

The frozen sections from the spinal cord are processed for the double immunofluorescence. In brief, these sections are blocked with 5% normal goat serum with 0.5% Triton X-100 in 0.5 M Tris for 1 h at room temperature and incubated with one of a mixture of primary antiserum at 4 °C overnight: (1) rabbit polyclonal TLR5 (1:100, 19810-1-AP, Proteintech Group, Inc., Chicago, IL, USA); (2) rabbit polyclonal pPKCα (1:200, 1195-1,Epitomics, Burlingame, CA, USA); (3) rabbit polyclonal pPKA RII (1:200, 1151-1, Epitomics, Burlingame, CA, USA); and (4) rabbit polyclonal MOR (1:200, RA10104, Neuromics, Edina, MN, USA) were mixed with mouse monoclonal SP (1:100, sc-58591, Santa Cruz Biotechnology; Santa Cruz, CA, USA), mouse monoclonal CGRP (1:100, sc-57053,Santa Cruz Biotechnology; Santa Cruz, CA, USA) or biotinylated IB4 (1:1000, L 2140, Sigma, UK) respectively. After rinsing in Tris, the sections are incubated with a mixture of secondary antibodies, that is, (1) Alexa Fluor^®^ 488-conjugated anti-rabbit IgG/Alexa Fluor^®^ 594-conjugated anti-mouse IgG (both from Invitrogen, Carlsbad, CA, USA) or (2) Alexa Fluor^®^ 488-conjugated anti-rabbit IgG/Phycoerythrin (PE) conjugated streptavidin (BioLegend, San Diego, CA, USA) for another hour. The slides were mounted with coverslips containing Vectashield Mounting Media (Vector Laboratories, Burlingame, CA, USA). The sections are photographed under a conventional epifluorescent microscope (Zeiss Axiophot, Carl Zeiss, Heidelberg, Germany) equipped with appropriate filters. The standard quantitation of dorsal horn areas wassetup and modified from the previous protocol.

### 4.7. Pharmacological Intervention

#### 4.7.1. Drugs

FLA-ST Ultrapure (purified flagellin from *Salmonella* Typhimurium, a TLR5 agonist) and the vehicle solution, endotoxin-free water, were purchased from InvivoGen (San Diego, CA, USA). In this experiment, it was applied at 0.1 μg, 0.3 μg and 0.9 μg in 10 μL vehicle solution separately, which was modified from the previous procedure of intraplantar administration with flagellin [19].

Go 6976 was purchased from Tocris Bioscience (Ellisville, MO, USA) and prepared in 0.1% DMSO in 0.1 M PB [57]. FLA-ST Ultrapure 0.9 μg (InvivoGen, San Diego, CA, USA) was used to combine with Go 6976, which separated into four subgroups (0 μM, 5 μM, 20 μM and 50 μM; *n* = 3 per subgroup) respectively. Likewise, DAMGO 25 μg (Tocris Bioscience, Ellisville, MO, USA) was used to combine with Go 6976 and also assigned into four subgroups separately.

#### 4.7.2. Intrathecal Administration

In order to reduce inflammatory responses caused by long-term intrathecal intubation, prepared FLA-ST Ultrapure or vehicle solution was administered through lumbar puncture by modifying an established protocol [58]. Briefly, under low-dose pentobarbital anesthesia (20 mg/kg, i.p.) with transiently ether anesthesia, the rat was placed in the prone position and a 4cm incision was made between the L4 and S1 spinal vertebrae. The paraspinal muscles were carefully dissected from their attachments and separated from the spinous processes of the vertebrae at the L4–S1 levels of the vertebral column. A 26-gauge needle connected to a 10-μL Hamilton syringe (model: 701, Hamilton Company, Reno, NV, USA) was inserted between the L4–L5 vertebrae at an angle of 60° horizontal to the subarachnoid space of the cauda equina. Access to the intrathecal space was confirmed by reflux of cerebro-spinal fluid. A volume of 10 μL containing FLA-ST Ultrapure or vehicle solution was slowly injected intrathecally in about 30 s.

#### 4.7.3. Behavioral Assessments

Before the evaluation, we first established the behavioral assessments in CCI group at POW 8 that were defined as post-injection hour (PIH) 0. Based on the different doses of FLA-ST Ultrapure, the rats in CCI group were separated into four subgroups (Vehicle, 0.1 μg FLA-ST Ultrapure, 0.3 μg FLA-ST Ultrapure and 0.9 μg FLA-ST Ultrapure groups; *n* = 3 per subgroup). After an intrathecal administration, each rat was measured in two periods: from 2 to 6 h (PIH 2, PIH 4 and PIH 6) and 24 h (PIH 24).

### 4.8. Statistical Analysis

Examiners were blinded to the grouping information when performing all the laboratory procedures of measurement and quantitation. Values are expressed as mean ± standard deviation using GraphPad Prism (GraphPad, San Diego, CA, USA). Statistical comparisons in behavioral assessments and dorsal horn area were made by a one-way repeated measures analysis of variance (ANOVA), followed by Bonferroni *post hoc* test, with the times as the within-subjects factor. Statistical comparisons in pharmacological intervention were made by the two-way ANOVA, followed by Bonferroni *post hoc* test, with the concentrations of a TLR5 agonist as between-subjects factors and time as the within-subjects factor. A probability value of less than 0.05 (*p* < 0.05) was considered statistically significant.

## 5. Conclusions

Based on the effectiveness of nerve decompression in the attenuations of CCI-induced pain hypersensitivity in Sprague-Dawley rats, we found that (1) increase of TLR5 expression significantly expressed in non-peptidergic terminals in dorsal horn; (2) FLA-ST Ultrapure (flagellin, a TLR5 agonist) had a role in the reduction of mechanical hyperalgesia by an intrathecal administration; (3) increase of MOR expression was in association with the activation of PKCα in dorsal horn; and (4) TLR5 contributed to MOR-mediated PKCα signaling for attenuating CCI-induced mechanical hyperalgesia. This provides a further look into the mechanism of cross-talk between TLR5 and MOR, which is critical for neuropathic pain management.

## Figures and Tables

**Figure 1 ijms-22-01891-f001:**
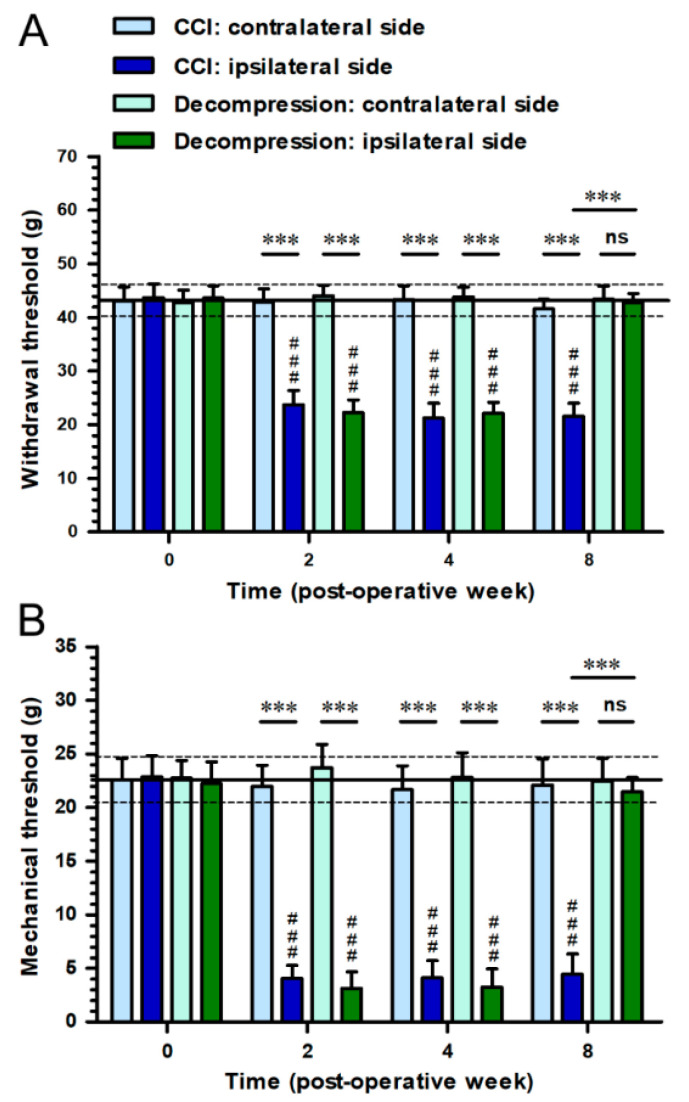
Effects of nerve decompression on pain hypersensitivity in the rodent model of chronic constriction injury (CCI). (**A**) The value of mechanical hyperalgesia, a safety pin-evoked withdrawal response, was measured as withdrawal threshold (g). (**B**) The degree of mechanical allodynia, a vonFrey filament-evoked withdrawal response, was recorded as mechanical threshold (g) at the following post-operative week. The temporal changes of pain hypersensitivity on ipsilateral sides in (**A**,**B**) CCI (dark blue bars) and (**C**,**D**) Decompression groups (dark green bars) were shown. Also, the comparable values on contralateral sides in CCI groups (light blue bars) and Decompression groups (light green bars) were exhibited at each time points in panels. All the values were expressed as mean ± standard deviation (SD) (*n* = 5 per time point). Statistical comparisons of withdrawal threshold and mechanical threshold were made by the one-way repeated measures analysis of variance (ANOVA), followed by Bonferroni *post hoc* test, with the times as the within-subjects factor. ### *p* < 0.001, indicated a significant difference. *** *p* < 0.001, indicated a significant difference and ns mean no significant difference at the same time point.

**Figure 2 ijms-22-01891-f002:**
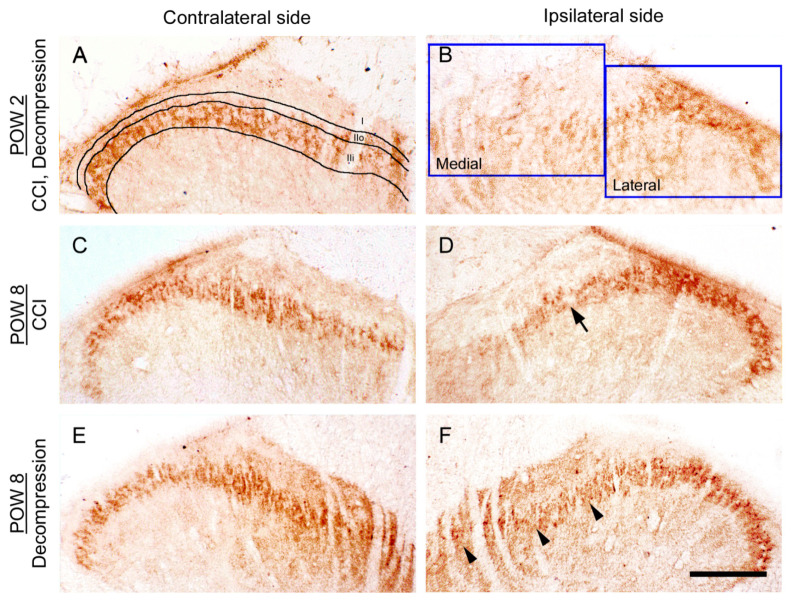
Influence of nerve decompression in the CCI-induced changes of Toll-like receptor 5 (TLR5) expression in dorsal horn. (**A**,**C**,**E**) Standard pattern of TLR5-immunoreactive (ir) expression was exposed on the contralateral sides of CCI and Decompression groups at post-operative week (POW)2 and POW 8. While on contralateral sides, TLR5-ir expression which showed a dotted appearance, mainly expressed in the inner part of laminae II. Related laminae of dorsal horn were demonstrated in (**A**), which involved lamina I (I), the outer part of lamina II (IIo) and the inner part of lamina II (IIi). (**B**,**D**,**F**) On ipsilateral sides, TLR5-ir expressions were exhibited in both groups. (**B**) At POW 2, CCI-induced reduction of TLR5-ir expression was revealed mainly in the medial portion of dorsal horn in both groups. (**D**,**F**) In Decompression group, more significant increase of TLR5-ir expression was observed at POW 8 (arrowheads in (**F**)), as compared with those in CCI group (arrow in (**D**)), especially in the corresponding lamina. Scale bar = 100 μm.

**Figure 3 ijms-22-01891-f003:**
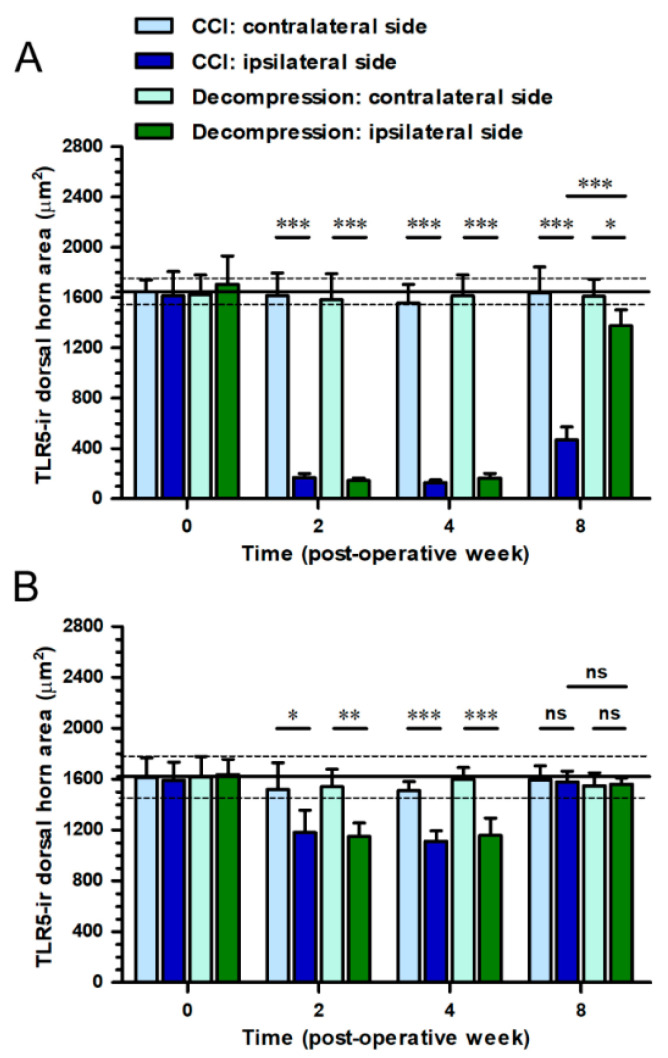
Quantitation of CCI-induced changes of TLR5 expression in dorsal horn after nerve decompression. In (**A**,**B**) CCI group (dark blue bars) andDecompression group (dark green bars), panels revealed the temporal changes of TLR5-ir expression on ipsilateral sides, which were quantified as the dorsal horn area (μm^2^) in the (**A**) medial and (**B**) lateral portions of dorsal horn respectively (quantitative areas were exhibited in panel B, Figure 2). The values on contralateral sides in CCI group (light blue bars) and Decompression group (light green bars) were also shown in panels for comparison. All the values were expressed as mean ± SD (*n* = 5 per time points). Student’s *t* test was applied to examine contralateral side vs. ipsilateral side at each time points. * *p* < 0.05, ** *p* < 0.01 and *** *p* < 0.001, indicated a significant difference and ns mean no significant difference.

**Figure 4 ijms-22-01891-f004:**
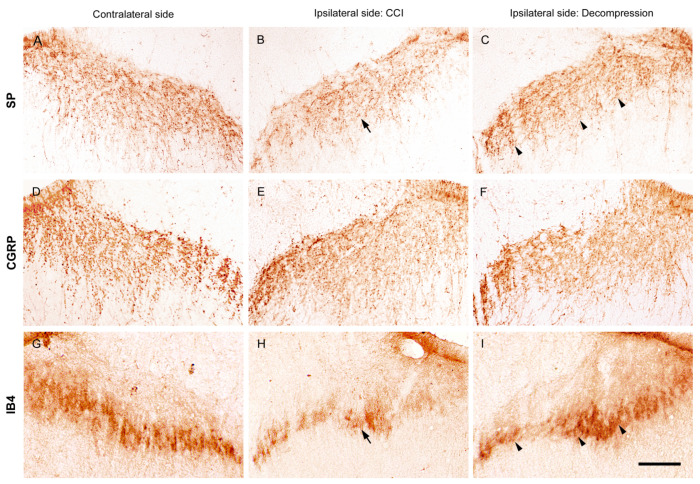
Effects of nerve decompression on the CCI-induced changes of peptidergic and non-peptidergic terminals in dorsal horn. (**A**,**D**,**G**) On contralateral sides, peptidergic ((**A**) substance P (SP) and (**D**) calcitonin gene-related peptide (CGRP))- and non-peptidergic ((**G**) isolectin B4 (IB4))-ir expressions in primary afferent terminals were shown in the medial portion of dorsal horn at POW 8. SP-ir expression formed beaded particles mostly in lamina I and extended to the outer part of lamina II. CGRP-ir expression exhibited as dense varicosities in lamina I, the outer part of lamina II and even projected to lamina V. In contrast, IB4-ir expression displayed an irregular structure and expressed generally in the inner part of lamina II. On ipsilateral sides, (**B**,**E**,**H**) CCI and (**C**,**F**,**I**) Decompression groups revealed that CCI-induced SP- and IB4-ir expressions rather than CGRP-ir expression had an obvious increase in their corresponding laminae after nerve decompression (arrowheads in (**C**,**I**)). Scale bar = 50 μm.

**Figure 5 ijms-22-01891-f005:**
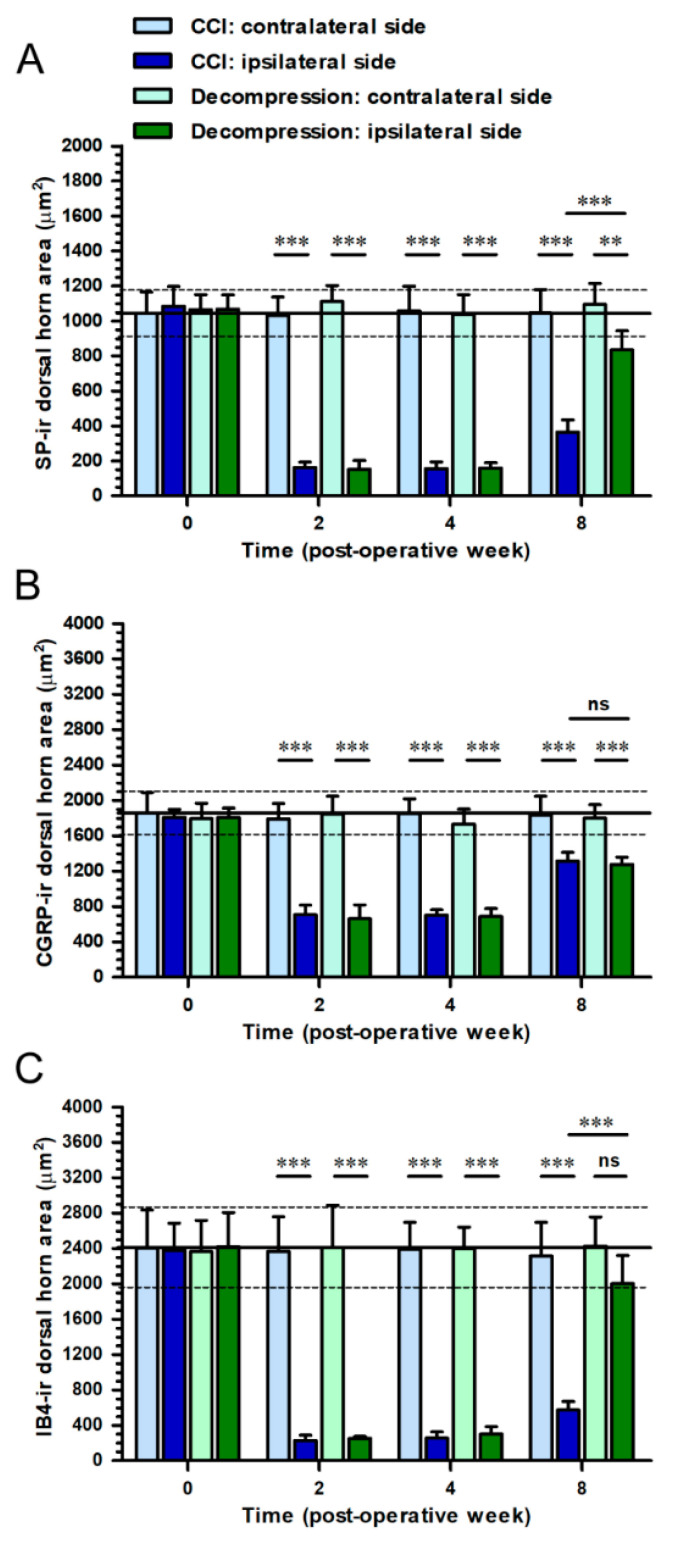
Quantitation of CCI-induced changes of peptidergic and non-peptidergic nerve terminals in dorsal horn after nerve decompression. In (**A**,**B**,**C**) CCI group (dark blue bars) and Decompression group (dark green bars), panels illustrated the temporal changes of (**A**) SP-, (**B**) CGRP- and (**C**) IB4-ir expressions on ipsilateral sides, which were quantified as the dorsal horn area (μm^2^) in the medial portion of dorsal horn. The values on contralateral sides in CCI group (light blue bars) and Decompression group (light green bars) were also shown in panels for comparison. All the values were expressed as mean ± SD (*n* = 5 per time points). Student’s *t* test was applied to examine contralateral side vs. ipsilateral side and CCI group vs. Decompression group at each time points. ** *p* < 0.01 and *** *p* < 0.001, indicated a significant difference and ns mean no significant difference.

**Figure 6 ijms-22-01891-f006:**
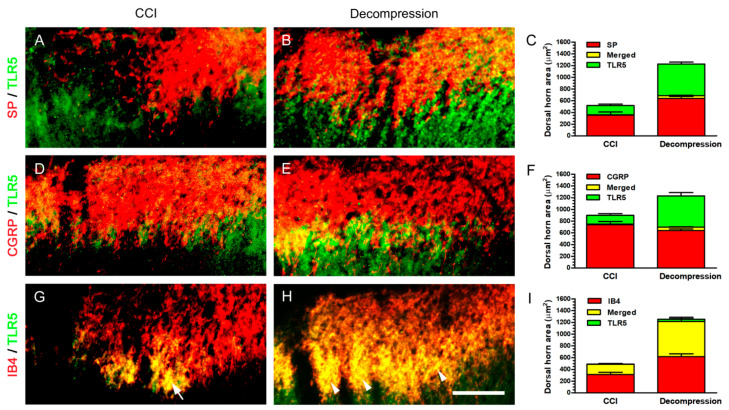
Identification of TLR5 expression in dorsal horn by double immunofluorescence.On ipsilateral sides, (**A**,**D**,**G**) CCI and (**B**,**E**,**H**) Decompression groups exhibited their related co-expressions in the medial portion of dorsal horn at POW 8. (**A**,**B**) The merged images illustrated the increase of TLR5-ir (green) expression in Decompression group almost co-expressed with SP-ir (red) expression in lamina I. (**D**,**E**) In the merged images, TLR5- (green) and CGRP-ir (red) expressions partially co-expressed, especially near the border between the outer and inner part of lamina II. (**E**,**F**) Importantly, TLR5-ir (green) expression mainly co-expressed with IB4-ir (red) expression in the inner part of lamina II (arrowheads in (**H**)). Also, the similar pattern of TLR5- and IB4-ir co-expression was demonstrated in CCI group (arrow in (**G**)). (**C**,**F**,**I**) Panels showed the quantitation of individual ir dorsal horn areas and their co-expressed areas. Scale bar = 40 μm.

**Figure 7 ijms-22-01891-f007:**
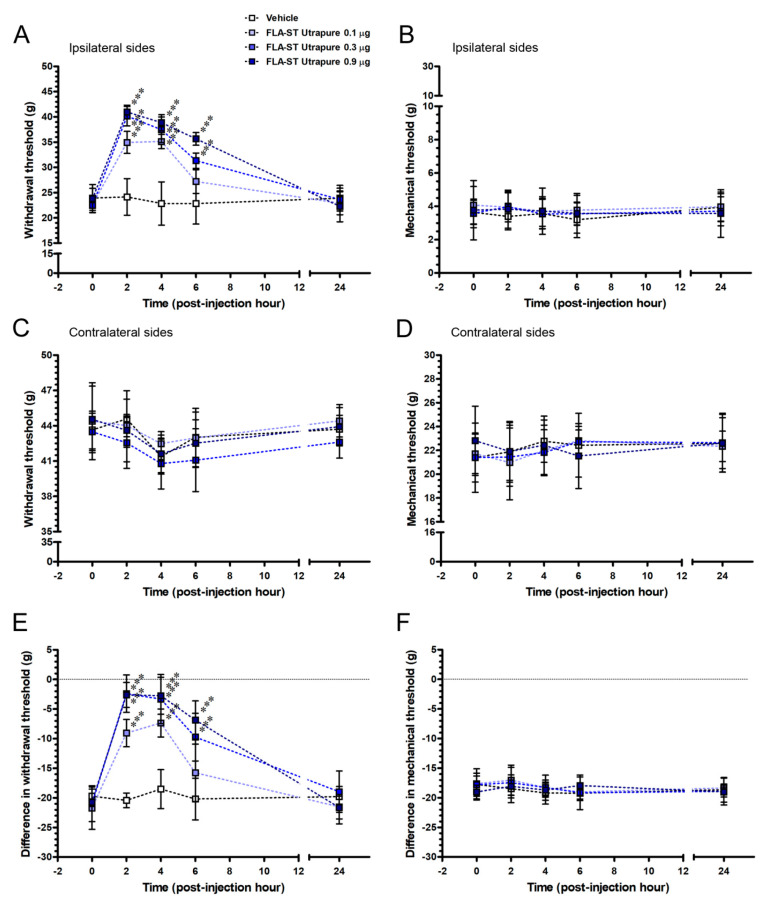
Role of FLA-ST Ultrapure (flagellin), a TLR5 agonist, in CCI-induced pain hypersensitivity by an intrathecal administration. (**A**,**C**) Mechanical hyperalgesia and (**B**,**D**) mechanical allodynia were evaluated in CCI rats at POW 8 and recorded as baseline values at post-injection hour 0. Temporal changes of pain hypersensitivity on (**A**,**B**) ipsilateral and (**C**,**D**) contralateral sides were evaluated in Vehicle and various FLA-ST Ultrapure groups. Temporal patterns of (**E**) difference in withdrawal threshold and (**F**) difference in mechanical threshold demonstrated that FLA-ST Ultrapure exerted an analgesic effect on CCI-induced mechanical hyperalgesia but not mechanical allodynia, in a dose-responsive manner. The withdrawal threshold and mechanical threshold were represented as mean ± SD. Vehicle group (white squares, *n* = 3) was used to demonstrate the effect of intrathecal administration on behavioral assessments. The properties of FLA-ST Ultrapure (flagellin) were tested at the concentration of 0.1 μg (light blue squares), 0.3 μg (blue squares) and 0.9 μg (deep blue squares) (*n* = 3 per group). Statistical comparisons in Vehicle and FLA-ST Ultrapure groups were made by the two-way ANOVA, followed by Bonferroni *post hoc* test, with the concentrations of FLA-ST Ultrapure as between-subjects factors and time as the within-subjects factor. *** *p* < 0.001, indicated a significant difference.

**Figure 8 ijms-22-01891-f008:**
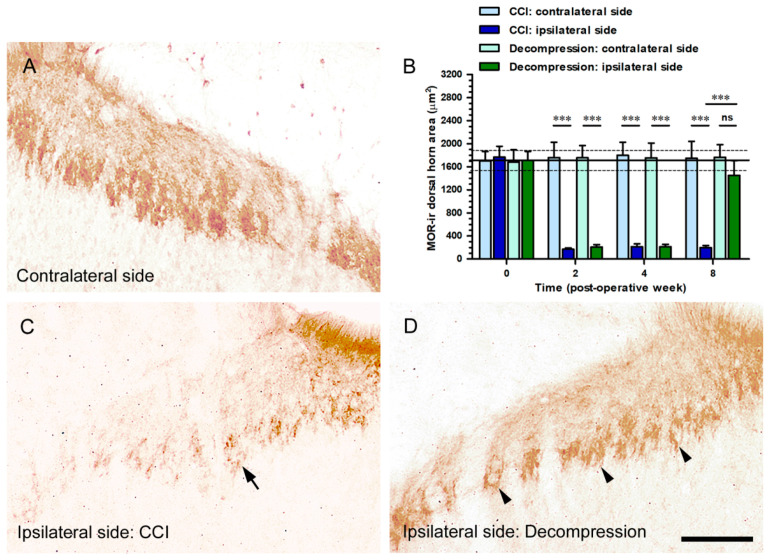
Effects of nerve decompression on the CCI-induced changes of mu-opioid receptor (MOR) in dorsal horn. (**A**) On contralateral sides, MOR expression was shown with abundant dense particles, which mainly expressed in laminae II. (**C**,**D**) At POW 8, MOR-ir expression in Decompression group (arrowheads in (**D**)) rather than those in CCI group (arrow in (**C**)) had a significant increase in their corresponding laminae. In (**B**) CCI group (dark blue bars) and Decompression group (dark green bars), panels illustrated the temporal changes of MOR-ir expression on contralateral and ipsilateral sides, which were quantified as the dorsal horn area (μm^2^) in the medial portion of dorsal horn. All the values were expressed as mean ± SD (*n* = 5 per time points). Student’s *t* test was applied to examine contralateral side vs. ipsilateral side and CCI group vs. Decompression group at each time points. *** *p* < 0.001, indicated a significant difference and ns mean no significant difference. Scale bar = 50 μm.

**Figure 9 ijms-22-01891-f009:**
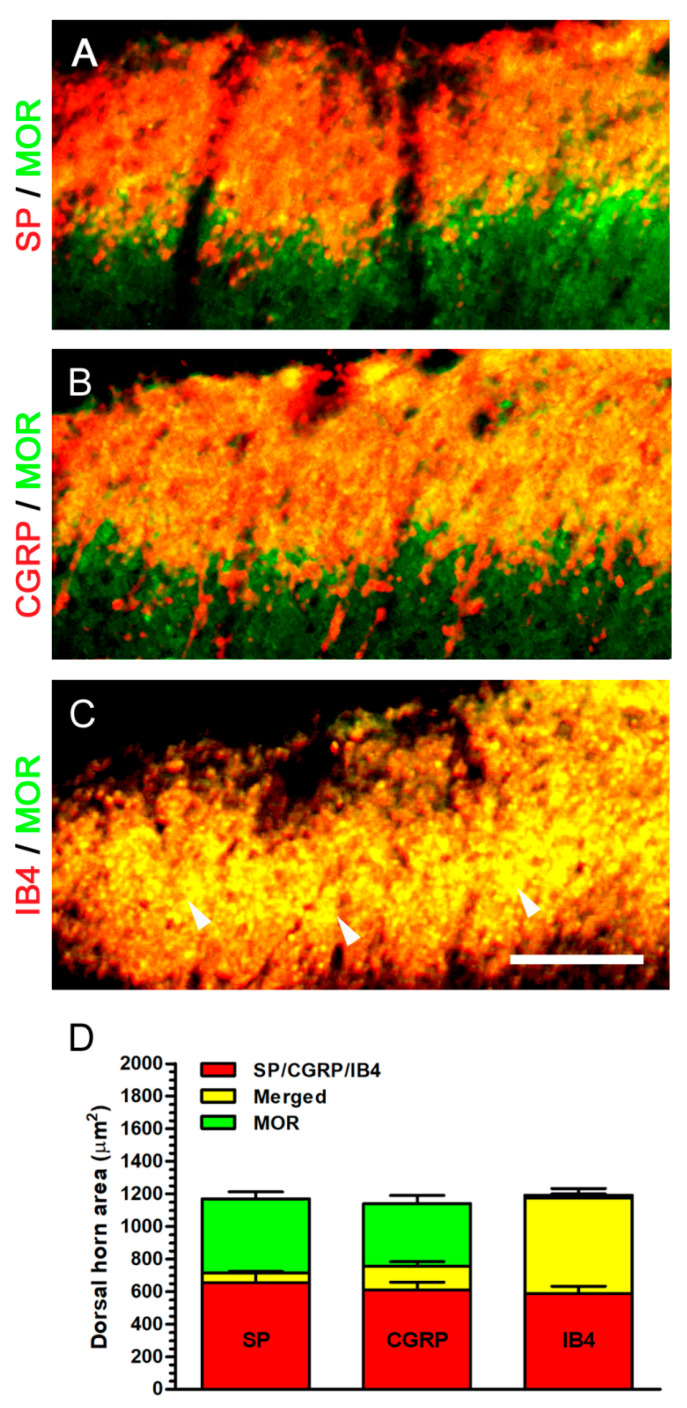
Identification of MOR expression in dorsal horn by double immunofluorescence. (**A**–**C**) In Decompression group, ipsilateral sides exhibited their related co-expressions with MOR-ir expressions in the medial portion of dorsal horn at POW 8. (**A**,**B**) The merged images illustrated the increase of MOR-ir (green) expression slightly co-expressed with SP- or CGRP-ir (red) expressions in lamina I and extended to the border between the outer and inner part of lamina II. (**C**) In the merged images, MOR- (green) and IB4-ir (red) expressions mostly co-expressed, especially in lamina II (arrowhead in (**C**)). (**D**) Panel showed the quantitation of individual ir dorsal horn areas and their co-expressed areas. Scale bar = 40 μm.

**Figure 10 ijms-22-01891-f010:**
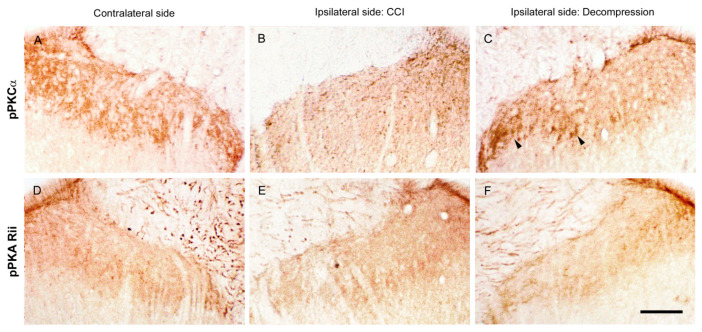
Influence of nerve decompression in the CCI-induced changes of protein kinase expressions in dorsal horn. (**A**,**D**) On contralateral sides, phospho-protein kinase Cα (pPKCα)- and phospho-protein kinase A RII (pPKA RII)-ir expressions were revealed in the medial portion of dorsal horn at POW 8. pPKCα-ir expression formed irregular particles which mostly terminated in lamina II. While pPKA RII-ir expression expressed as loose dotted appearances in the equivalent lamina. On ipsilateral sides, (**B**,**E**) CCI and (**C**,**F**) Decompression groups exposed the observations that CCI-induced pPKCα-ir expression rather than pPKA RII-ir expression noticeably increased in their corresponding lamina after nerve decompression (arrowheads in (**C**)). Scale bar = 50 μm.

**Figure 11 ijms-22-01891-f011:**
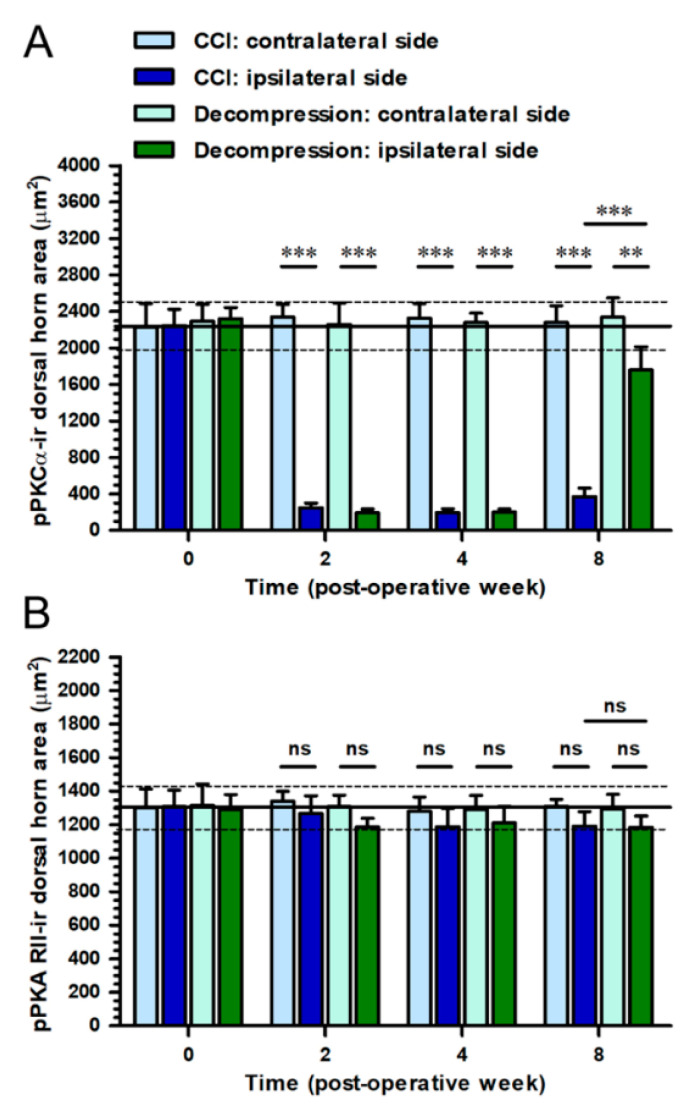
Quantitation of CCI-induced changes of protein kinase expressions after nerve decompression. In (**A**,**B**) CCI group (dark blue bars) and Decompression group (dark green bars), panels illustrated the temporal changes of (**A**) pPKCα-and (**B**) pPKA RII-ir expressions on ipsilateral sides, which were quantified as the dorsal horn area (μm^2^) in the medial portion of dorsal horn. The values on contralateral sides in CCI group (light blue bars) and Decompression group (light green bars) were also shown in panels for comparison. All the values were expressed as mean ± SD (*n* = 5 per time points). Student’s *t* test was applied to examine contralateral side vs. ipsilateral side and CCI group vs. Decompression group at each time points. ** *p* < 0.01 and *** *p* < 0.001, indicated a significant difference and ns mean no significant difference.

**Figure 12 ijms-22-01891-f012:**
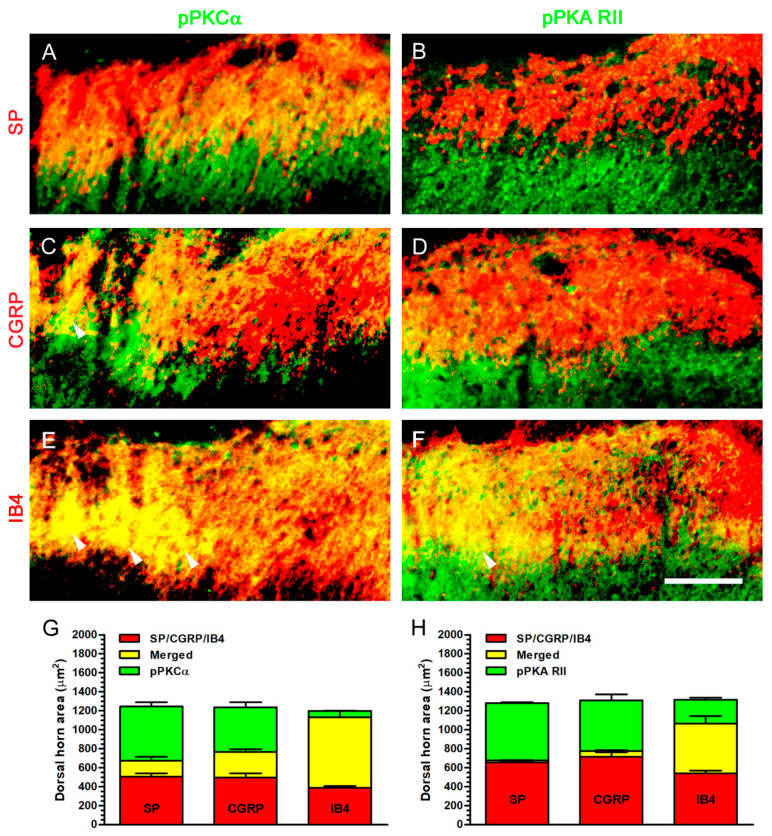
Identification of protein kinase expressions in dorsal horn by double immunofluorescence after nerve decompression. On ipsilateral sides, Decompression group exhibited (**A**,**C**,**E**) pPKCα- and (**B**,**D**,**F**) pPKA RII-ir expressions with their related co-expressions in the medial portion of dorsal horn at POW 8. (**A**,**C**) The merged images illustrated that the increases of pPKCα-ir (green) expression partially co-expressed with SP- or CGRP-ir (red) expressions around the border between the outer and inner part of lamina II. (**E**) In the merged images, pPKCα- (green) and IB4-ir (red) expressions closely co-expressed, especially in the inner part of lamina II (arrowheads). (**B**,**D**) pPKA RII-ir (green) expression almost not co-expressed with SP- or CGRP-ir (red) expressions. (**F**) pPKA RII-ir (green) expression showed more significant co-expression with IB4-ir expression (arrowhead). (**G**,**H**) Panels showed the quantitation of individual ir dorsal horn areas and their co-expressed areas. Scale bar = 40 μm.

**Figure 13 ijms-22-01891-f013:**
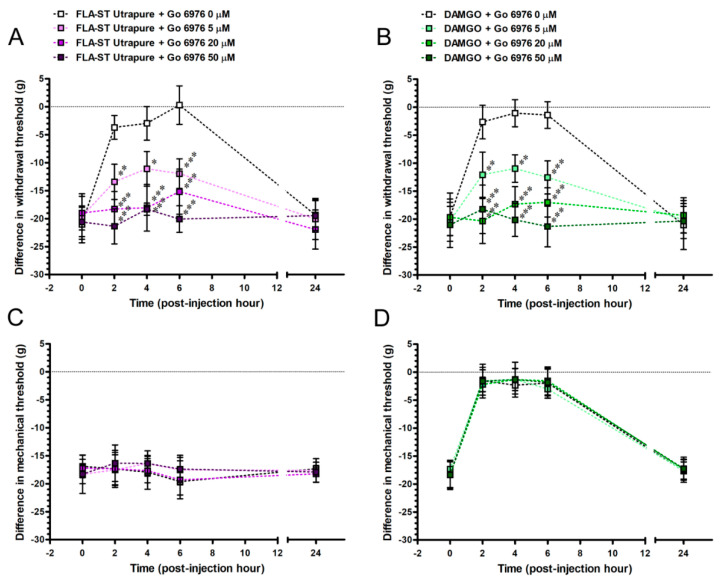
Role of Go 6976, a protein kinase C (PKC)inhibitor, in TLR5- and MOR-mediated attenuation of pain hypersensitivity by an intrathecal administration. (**A**,**B**) Mechanical hyperalgesia and (**C**,**D**) mechanical allodynia were evaluated in CCI rats at POW 8 and recorded as baseline values at post-injection hour 0. Temporal changes of (**A**,**C**) FLA-ST Ultrapure- and (**B**,**D**) DAMGO-mediated attenuation of pain hypersensitivity were affected by an intrathecal Go 6976 administration. (**A**,**B**) Time-based modes of difference in withdrawal threshold demonstrated that Go 6976 exerted a positive effect on the re-induction of FLA-ST Ultrapure- and DAMGO-mediated attenuation of mechanical hyperalgesia, in a dose-responsive manner. The difference in withdrawal threshold and difference in mechanical threshold were represented as mean ± SD. FLA-ST Ultrapure + Go 6976 0 μM group or DAMGO + Go 6976 0 μM group (white squares, *n* = 3) was used to demonstrate the effect of intrathecal administration on behavioral assessments. The properties of Go 6976 were tested at the concentration of 5 μM (light purple squares in (**A**,**C**), light green squares in (**B**,**D**)), 20 μM (purple squares in (**A**,**C**), green squares in (**B**,**D**)) and 50 μM (deep purple squares in (**A**,**C**), deep green squares in (**B**,**D**)) (*n* = 3 per group). Statistical comparisons were made by the two-way ANOVA, followed by Bonferroni *post hoc* test, with the concentrations of Go 6976 as between-subjects factors and time as the within-subjects factor. * *p* < 0.05, ** *p* < 0.01 and *** *p* < 0.001, indicated a significant difference.

## Data Availability

Data available on request due to restrictions e.g., privacy or ethical. The data presented in this study are available on request from the corresponding author. The data are not publicly available due to publicated methods of quantification in the materials and methods.

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
