# Peer review of "Cross-Talk of Toll-Like Receptor 5 and Mu-Opioid Receptor Attenuates Chronic Constriction Injury-Induced Mechanical Hyperalgesia through a Protein Kinase C Alpha-Dependent Signaling"

_ijms, 2021, doi:10.3390/ijms22041891_

Round 1
Reviewer 1 Report
This manuscript describes CCI-induced hyperalgesia and its attenuation by decompression through PKC alpha-dependent signaling and suggests TLR5/MOR cross-talk in the process. Overall, this manuscript is written OK, together with enough data to support the conclusions, although n=3/subgroup seems weak for statistic value. Therefore, the reviewer recommends this manuscript published in this journal after minor revisions. Please see below.
- Figures 1 and 3: combine A and C, and B and D; delete E and F
- Typos and overlapped abbreviations: ex. line 136, the full name of TLR5; line 645; line 680; etc
- The conclusion part needs to be polished. The sentences are hard to understand and need to be rewritten.
Author Response
ijms-1053976_Reviewer1
Comments and Suggestions for Authors
This manuscript describes CCI-induced hyperalgesia and its attenuation by decompression through PKC alpha-dependent signaling and suggests TLR5/MOR cross-talk in the process. Overall, this manuscript is written OK, together with enough data to support the conclusions, although n=3/subgroup seems weak for statistic value. Therefore, the reviewer recommends this manuscript published in this journal after minor revisions. Please see below.
1. Figures 1 and 3: combine A and C, and B and D; delete E and F
2. Typos and overlapped abbreviations: ex. line 136, the full name of TLR5; line 645; line 680; etc
3. The conclusion part needs to be polished. The sentences are hard to understand and need to be rewritten.
Response:
Thanks to the comments of reviewer #1.
Point-to-point specific responses to comments are described in the following pages.
1. We have done and re-arranged figure panels (Figure 1, 3, 5, 8, and 11) to meet a requirement in revised manuscript.
2. We have noticed these typing errors and corrected. Our manuscript has been edited by the scientific proofreading.
3. We have rewritten the sentences in the conclusion to meet a requirement in revised manuscript.
Reviewer 2 Report
The manuscript of Ching Chang et al. investigated whether is a possible cross talk between Toll-like Receptor 5 (TLR-5) and µ opioid receptors in the decompression induced attenuation of mechanical hyperalgesia produced by neuropathic pain.
The introduction is well written, the topic is well presented. However, the aim of the study was not described. The main questions are missing and also not known why the specific experiments were performed.
After the introduction the reader gets into the middle of the results without knowing why they were done.
The representation of the figures is very poor. It is difficult to understand which figure represents which result.
In the case of the measurement of the pain reaction the statistic is not correct; mechanical hyperalgesia and mechanical allodynia were measured several times on the same animal. In this case a repeated measures ANOVA is correct, not the T-test.
It is difficult to recognize the signals on the images that represent the immunohistochemical staining, therefore it is hard to follow their quantification. This is also true for the dorsal horn index. It is not clear how this index was calculated.
The images supposed to prove the different colocalizations (TLR-5 – SP, TLR-5 – CGRP, TLR-5 – IB4, SP – MOR, CGRP – MOR, IB4 - MOR) were not quantified at all.
The discussion is wordy but it is hard to see the connection between the statements and the results.
On the base of the poor presentation of the results, it is difficult to believe the conclusion drawn. Therefore, I suggest a substantial revision of the manuscript before it could be published.
Author Response
ijms-1053976_Reviewer2
Comments and Suggestions for Authors
The manuscript of Ching Chang et al. investigated whether is a possible cross talk between Toll-like Receptor 5 (TLR-5) and µ opioid receptors in the decompression induced attenuation of mechanical hyperalgesia produced by neuropathic pain.
The introduction is well written, the topic is well presented. However, the aim of the study was not described. The main questions are missing and also not known why the specific experiments were performed.
After the introduction the reader gets into the middle of the results without knowing why they were done.
The representation of the figures is very poor. It is difficult to understand which figure represents which result.
In the case of the measurement of the pain reaction the statistic is not correct; mechanical hyperalgesia and mechanical allodynia were measured several times on the same animal. In this case a repeated measures ANOVA is correct, not the T-test.
It is difficult to recognize the signals on the images that represent the immunohistochemicalstaining, therefore it is hard to follow their quantification. This is also true for the dorsal horn index. It is not clear how this index was calculated.
The images supposed to prove the different colocalizations (TLR-5 – SP, TLR-5 – CGRP, TLR-5 – IB4, SP – MOR, CGRP – MOR, IB4 - MOR) were not quantified at all.
The discussion is wordy but it is hard to see the connection between the statements and the results.
On the base of the poor presentation of the results, it is difficult to believe the conclusion drawn. Therefore, I suggest a substantial revision of the manuscript before it could be published.
Response:
Thanks to the comments of reviewer #2.
Point-to-point specific responses to comments are described in the following pages.
1. Thanks for the suggestions of reviewer #2, we have added the aim of the study and list the main questions in the section of the introduction to help understanding the following statement in the results of the revised manuscript.
2. In order to get rid of redundant quantifying figures that would distract from reading data, we decreased the quantity of figures. Besides, we applied absolute quantifying data to demonstrate the difference.
3. We agreed the ANOVA measurement in analyzing pain hypersensitivity. Data has been used to re-analyze and present in the revised manuscript.
4. We have decided to delete the related figures that included dorsal horn index in the revised manuscript. In addition, photoshop software is used to calculate the immunoreactive area automatically in our current study.
5. We have consolidated the figures of double immunofluorescence with their quantification, e.g., Figure 3, Figure 6, and Figure 9 in the revised manuscript.
6. For more accurate statements, we have polished the discussion to meet a requirement in the revised manuscript.
Round 2
Reviewer 2 Report
The manuscript improved substantially. The revised form is suitable for publication.